# Impact of Dietary Fructose and High Salt Diet: Are Preclinical Studies Relevant to Asian Societies?

**DOI:** 10.3390/nu14122515

**Published:** 2022-06-17

**Authors:** Ban Hock Khor, Dragana Komnenov, Noreen F. Rossi

**Affiliations:** 1Faculty of Food Science and Nutrition, Universiti Malaysia Sabah, Kota Kinabalu 88400, Malaysia; khorbanhock@gmail.com; 2Department of Internal Medicine, Wayne State University, Detroit, MI 48201, USA; dkomneno@wayne.edu; 3Department of Physiology, Wayne State University, Detroit, MI 48201, USA; 4Division of Research, John D. Dingell VA Medical Center, Detroit, MI 38201, USA

**Keywords:** Asian population, blood pressure, cardiovascular risk, fructose, hypertension, renin–angiotensin–aldosterone system, salt, sodium chloride, sympathetic nervous system

## Abstract

Fructose consumption, especially in food additives and sugar-sweetened beverages, has gained increasing attention due to its potential association with obesity and metabolic syndrome. The relationship between fructose and a high-salt diet, leading to hypertension and other deleterious cardiovascular parameters, has also become more evident, especially in preclinical studies. However, these studies have been modeled primarily on Western diets. The purpose of this review is to evaluate the dietary habits of individuals from China, Japan, and Korea, in light of the existing preclinical studies, to assess the potential relevance of existing data to East Asian societies. This review is not intended to be exhaustive, but rather to highlight the similarities and differences that should be considered in future preclinical, clinical, and epidemiologic studies regarding the impact of dietary fructose and salt on blood pressure and cardiovascular health worldwide.

## 1. Introduction

High blood pressure is a major risk factor for cardiovascular morbidity and mortality. The contribution of modifiable risk factors to the development and progression of hypertension has been extensively discussed in the literature, with one of the main contributors being diet [1,2,3,4,5]. Indeed, the presence of high salt, sugar, and fat—alone or in different permutations in the diet—has been shown to contribute to the development of cardiovascular, renal, and endocrine disorders [6,7,8,9,10]. Several preclinical studies have been conducted mimicking a Western-type diet [11,12,13,14,15,16,17,18,19]; however, there is a paucity of preclinical data on diets high in salt and moderate in sugar-sweetened beverages (SSB) consumed by individuals living in Asian countries, such as Japan, the Republic of Korea, the People’s Republic of China, and Taiwan. The aim of this review is to critically assess whether existing preclinical findings regarding fructose and salt consumption may be relevant to East Asian populations.

## 2. Materials and Methods

We searched PubMed (https://www.ncbi.nlm.nih.gov/pubmed/ (accessed on 2 June 2022)), the Web of Science Core Collection (https://www.library.ethz.ch/en/Resources/Databases/Web-of-Science-Core-Collection (accessed on 25 April 2022)), and the Cochrane Registry (http://www.cochranelibrary.com/about/central-landing-page.html (accessed on 25 April 2022)) from January 1980 to April 2022. The following search terms were used: fructose and blood pressure or hypertension; fructose and sodium; fructose and kidney; fructose and salt; fructose and insulin resistance; high-salt diet; fructose and salt and sympathetic nervous system; fructose and renal transporters; fructose and intestinal absorption; Asian population; fructose and China; fructose and Japan; fructose and Korea; sugar-sweetened beverages and China; sugar-sweetened beverages and Japan; sugar-sweetened beverages and Korea; salt or sodium diet and China; salt or sodium diet and Japan; and salt or sodium diet and Korea. Relevant human and animal studies that addressed fructose and salt consumption in East Asian populations were included. In addition, reviews regarding consumption of these nutrients by Western countries, as well as animal studies—primarily in rodents—on fructose and a high-salt diet, were used for comparison. The review is not intended to be exhaustive, but rather to be a critical appraisal and comparison of model diets and their relevancy to East Asian vs. Western populations. Although there were more than 20,000 items found regarding sodium and 1372 reports regarding fructose in East Asia, only 78 references were found when searching human studies on fructose AND sodium with respect to either China, Korea, or Japan. This highlights the paucity of articles regarding the combined dietary intakes in East Asian populations.

This review focuses on the impact of fructose and salt consumption on the development of hypertension, independent or preceding the full expression of metabolic syndrome, which may lead to heightened cardiovascular risk. The influence of fructose on metabolic syndrome is well recognized and reviewed in detail in several recent reviews [20,21,22,23]. 

## 3. Results

### 3.1. Susceptibility to Hypertension: Genetics

The factors influencing one’s susceptibility to high blood pressure may be related to genetic [24] or lifestyle characteristics [10]. Several genome-wide association studies (GWASs) revealed more than 200 genetic loci related to hypertension development in individuals of European ancestry [24,25,26,27,28]. Similar studies, albeit on a smaller scale, have been conducted in East Asian individuals [29,30,31]. Up until 2016, data regarding Asian individuals were more limited, which is important given that allele frequencies appear to be less common across ethnic and racial groups than within groups [24]. More recently, a large GWAS was conducted in East Asian populations and compared to Europeans, which revealed 19 new genetic loci and proposed a common ancestry-specific variant association model in these two cohorts [31]. Nevertheless, hypertension is more prevalent in people of East Asian ancestry compared to European ancestry. Thus, although some differences can be ascribed to genetic susceptibilities, one can argue that lifestyle factors, particularly the modifiable ones such as diet, have gained less attention. The higher hypertension prevalence in Asian populations may, in part, explain the higher morbidity and mortality rates due to stroke rather than coronary heart disease in these individuals, whereas the converse is true in Western countries [32]. Alternatively, the impact of hypertension on cerebrovascular versus cardiovascular systems may vary among different ethnicities, due to other genetic, epigenetic, environmental, and lifestyle factors.

### 3.2. Salt-Sensitive Blood Pressure

High salt intake has been described as one of the most prominent environmental factors contributing to hypertension [33,34,35]. Table 1 shows the reported salt intake for several large studies in East Asian populations. Individuals may display normal blood pressure but respond to a high-salt diet with an increase in blood pressure, termed salt-sensitive blood pressure (SSBP) [36]. Interestingly, SSBP is an independent risk factor for cardiovascular disease, whether the individual is normotensive or hypertensive at baseline [37]. The causal association between high salt intake and blood pressure has been unequivocally demonstrated by the INTERMAP (International Study of Macro- and Micro-nutrients and Blood Pressure) study, which included 52 study sites distributed throughout 32 countries, including Asian sites [38]. A recent cohort study conducted in Northern China found that among 2057 participants, SSBP prevalence is the same between hypertensive (28.5%) and normotensive (28.2%) groups, with daily salt intake being higher in the former (8.97 g/day) compared to the latter (6.87 g/day) [39]. Notably, these daily intakes are well above the upper limit of 1500 mg of sodium/day recommended by the American Heart Association, which equates to about 3.8 g of salt/day [40]. In addition to dietary factors (i.e., sodium intake), specific gene loci have been associated with SSBP. These occur in the genes encoding the proteins involved in pathways regulating salt balance. Among these are genes encoding for angiotensinogen [41], α-adducin [42], aldosterone synthase [43], and the G-protein coupled to the Na^+^-H^+^ exchanger [44]. One study compared the frequency of these alleles between Japanese and Caucasian cohorts, and found that they were all significantly higher in the former group [45]. The GenSalt study also identified genetic associations of SSBP with potassium-selective ion channels (K_ir_), cyclic GMP-dependent protein kinase genes, and the epithelial sodium channel (ENaC) among participants in rural China [46,47,48]. An association between SSBP and single nucleotide polymorphisms (SNPs) has also been found in Korean individuals who have emigrated to the United States [49]. 

Additional gene polymorphisms associated with hypertension have been reported in Asians [61], but these are not as clearly related to salt sensitivity. Given such genetic susceptibility to SSBP and the enhanced cardiovascular risk imparted by hypertension, it is reasonable that strategies to reduce sodium intake in East Asian populations would be worthwhile. 

The guidelines for salt intake by the Japanese Society of Hypertension [62], the Dietary Guidelines for Chinese Residents [63], and the Korean Society of Hypertension [64] all recommend <6 g/d of dietary salt or 2.4 g/d of dietary sodium, which is comparable to the 2.3 g/d of sodium intake (equivalent to 5.85 g/d of salt) endorsed by the American College of Cardiology and American Heart Association [65]. Since many of the studies regarding dietary salt intake and various associated clinical parameters, including hypertension, depend on questionnaires or recall, it is noteworthy that verification by urinary sodium collection from the INTERMAP study showed agreement with the dietary recall of salt intake within 2.9 and 3.9 mmol/d for individuals from Japan, the United Kingdom, and the United States. In contrast, excretion of salt for the Chinese cohort averaged 54.0 mmol/d (~3.2 g/d of salt) greater than that reported by dietary recall. The estimated misclassification of salt intake ranged from 25.4% for the United Kingdom to 58.6% for China [57]. Thus, the reliability of dietary recall and questionnaires for salt intake (as well as other nutrients) should be viewed with some caution. 

### 3.3. Fructose Intake, Blood Pressure, and Cardiovascular Risk

In Western societies, fructose ingestion has been correlated with higher blood pressures [66,67]; an interaction with sodium intake has also been demonstrated [6,68,69,70]. Despite its low glycemic index, the association between fructose, obesity and cardiovascular disease has been strong enough that Segal et al. [71] proposed adoption of a fructose index. The majority of reports address fructose in relation to obesity, diabetes, or metabolic syndrome. The results from the large Western CARDIA (Coronary Artery Risk Development in Young Adults; *n* = 240,508) cohort indicated that the risk of hypertension with consumption of SSB is 12% higher, even when controlled for sex, age, race, BMI and smoking history [72]. Sucrose, a disaccharide comprising fructose and glucose, can also induce elevations in blood pressure when paired with high salt intake [73]. The deleterious effects of fructose related primarily to the levels of fructose that can be achieved in the systemic and portal bloodstream. Notably, greater amounts of dietary sucrose are required to achieve the levels of fructose that result from consumption of high-fructose corn syrup [74]. Moreover, in a report by the Mayo Clinic, evaluating 24 prospective cohort studies comprising 624,128 unique individuals, total sugars and fructose were associated with cardiovascular mortality, but not incidence. Sucrose was not associated with either cardiovascular incidence or mortality [75]; hence, the focus of this review was on fructose and salt ingestion. 

In general, the average fructose intake by individuals in East Asia tended to be lower (Table 2) than that observed in Western nations [8,62,66], where values averaged 49 g/d in adults, but can be as high as 75 g/d—especially in adolescents. The fructose in Western diets is ingested mostly in the form of high-fructose corn syrup found in beverages and food additives [76]. A superficial inspection of the publications regarding SSB consumption in Asian countries suggests that intake has increased over the last two decades, with the highest intakes in adolescents and males (Table 3), which mirrors the epidemiologic findings in Western societies [8,76]. The lack of a standard for reporting consumption of fructose and SSB, such as absolute caloric intake per day vs. percent of total caloric intake, makes comparisons among studies difficult. The problem is compounded by different definitions that constitute a serving of SSB, as well as variability in the caloric value (anywhere from 44 to 108 kcal/250 mL) of different SSB. 

There is a paucity of data regarding the impact, if any, of fructose intake, either alone or together with dietary sodium, on blood pressure exclusively in Asian populations. The INTERMAP study [69,85] that examined the relationship of blood pressure as a major risk factor for cardiovascular disease included individuals from Japan, the People’s Republic of China, the United Kingdom, and the United States. Unfortunately, the data on SSB (that included high-fructose corn syrup) showing a correlation with higher risk of cardiovascular events included samples from the United Kingdom and the United States only [86]. 

One randomized, double-blind, cross-over trial enrolled 18 healthy young Chinese adults of both sexes and examined the effect of drinking a 25% glucose or 25% fructose solution. Systolic blood pressure was significantly higher, as were aldose reductase and uric acid levels, whereas serum nitric oxide levels were lower with fructose at both 1 h and 3 h post-ingestion. The impact on diastolic blood pressure was similar between fructose and glucose trials. Consistent with its low glycemic index, fructose did not increase serum glucose or insulin levels [101]. Comparable large clinical trials or epidemiological studies regarding the impact of fructose, either with or without salt modulation, on blood pressure in Asian populations are lacking and are very much needed.

### 3.4. Preclinical Data on Dietary Fructose, High Salt, and Blood Pressure

Rodent models have used fructose diets providing >60% of total daily caloric intake to induce insulin-resistant hypertension [11,71,72]. More recent studies have modified the fructose content to 20% in order to better simulate the typical intake of the upper quintile of Western diets [15]. In short-term experiments (2–3 weeks), rats that were fed 20% fructose *w*/*w* in drinking water did not display elevated blood pressure compared with control rats given water or 20% glucose. However, when paired with a high-salt diet (4% NaCl), systolic blood pressure by tail-cuff measurements was significantly elevated (140 ± 2 mmHg) compared with the controls (122 ± 1 mmHg) or with rats fed fructose alone (128 ± 1 mmHg) [15]. Hypertension in rats fed fructose plus high-salt diets has also been reported using telemetry, albeit with more modest increases in blood pressure [102,103,104]. The fructose intake by Asian populations is generally less than that by individuals from Europe or America, but more recent findings suggest a troubling rise, particularly in SSB consumption by adolescents [91,94,95,98]. This would suggest that preclinical data regarding fructose, either with or without high-salt intake, will be more relevant to societies worldwide. 

Fructose is absorbed predominantly via GLUT5, which has a low affinity for glucose but high affinity for fructose in the jejunum. When large amounts of fructose are ingested and absorbed, the conversion of fructose to pyruvate by the liver is exceeded. The excess fructose enters the circulation, where fructose can then both enhance intestinal sodium absorption and increase renal tubule sodium reabsorption, leading to extracellular volume expansion [16,19,70,102,103,104]. The putative anion transporter-1 (PAT1, also known as SCL26A6 in humans) is a chloride/base exchanger on the apical surface of intestinal cells. Jejunal PAT1 colocalizes with GLUT5; PAT1 increases with fructose feeding and is coupled with the intestinal Na/H exchanger-3 (NHE3) [102,105,106,107]. Consequently, sodium and chloride absorption by the gut are increased. Thus, when delivered concurrently with fructose, a rise in dietary sodium does not result in higher fecal sodium excretion in rats [15]. The maintenance of sodium homeostasis, therefore, depends upon renal excretion. 

Urinary excretion of sodium was impaired in fructose-fed rats (Figure 1). Cumulative sodium balance is positive, which was not observed in glucose-fed rats [15,108]. Proximal tubule sodium reabsorption in vitro in a porcine cell line was enhanced in the presence of fructose [109]. Likewise, the activity of a proximal tubule Na/H exchanger, NHE3, isolated from rats fed 20% fructose in drinking water was augmented in a PKC-dependent manner when glucose was replaced with fructose in the test bath. Significantly, systolic blood pressure was 13 mmHg higher in the fructose-fed rats on a high-salt diet vs. rats fed a high-salt diet alone [110] or rats fed glucose and a high-salt diet [69]. Additional studies demonstrated that NHE3 activity was further potentiated by angiotensin II (Ang II) [16,104]. Although Na,K-ATPase activity did not appear to be altered by fructose [110], the proximal tubule expression of sodium-linked glucose transporters that are capable of co-transporting sodium and fructose, SGLT4 and SGLT5, was increased in fructose-fed rats, leading to greater reabsorption of both substances and decreased urinary sodium excretion [16]. In addition to augmentation of proximal tubule sodium reabsorptive processes, fructose feeding also appeared to increase the expression of sodium, potassium, and 2 chloride co-transporter (NKCC2) in the thick ascending limb of Henle [19]. 

Fructose consumption impacts the renin–angiotensin–aldosterone system. Expansion of extracellular volume and elevated blood pressure are capable of independently suppressing renin secretion; however, the suppression of plasma renin activity (PRA) was reduced in rats fed 20% fructose and 4% salt [15]. Notably, the sustained levels of PRA and Ang II resulted from renal sympathetic nerve activity, which was also increased in rats fed fructose and a high-salt diet. Consistent with this mechanism, renal cryo-denervation produced a decrease in PRA and Ang II in this model [111]. Importantly, sympathetic inputs to the renal tubules also stimulated sodium reabsorption all along the nephron [112]. The potentiation of sodium transport by Ang II, along with the failure to optimally suppress PRA and Ang II, contributed to further positive sodium balance. Furthermore, the direct effect of Ang II to induce vasoconstriction, combined with an expanded vascular volume and impaired baroreflex sensitivity [113], resulted in hypertension. 

Excess fructose consumption also contributed to the development of insulin resistance in preclinical models, despite the absence of frank hyperglycemia [108,114]. The mechanisms relating to fructose, insulin resistance, and hypertension have been elegantly reviewed by Tran et al. [11] and Xu [115]. In general, hyperinsulinemia and hyperleptinemia induced by fructose promoted sympathoexcitation, endothelial dysfunction, and metabolic derangements that led to oxidative stress, reduced nitric oxide, and the stimulation of cytokines and immune mechanisms [116,117,118]. Together, these conspire to decrease arterial compliance and increase vasoconstriction. Both insulin resistance and aortic stiffness can be ameliorated by free radical scavengers [114] and renal nerve ablation [111]. 

## 4. Conclusions

This overview of preclinical and clinical data is not meant to be exhaustive. Rather, we have highlighted some of the basic studies, mostly performed in rodents, that have demonstrated the potential deleterious impact that consumption of substantial amounts of fructose with a high-salt diet can exert on blood pressure, the sympathetic nervous system, and cardiovascular parameters. Clinical studies have clearly emphasized the association of hypertension, sympathoexcitation, and arterial stiffness with increased cardiovascular morbidity and mortality. In general, individuals living in Japan, Korea, and China tend to have higher salt intake and lower fructose consumption compared with the Western populations that the preclinical studies were designed to model. Additional preclinical investigations that model Asian dietary patterns, and more rigorous clinical and epidemiological studies that take into consideration the genetic and environmental nuances of those societies, will be needed.

## Figures and Tables

**Figure 1 nutrients-14-02515-f001:**
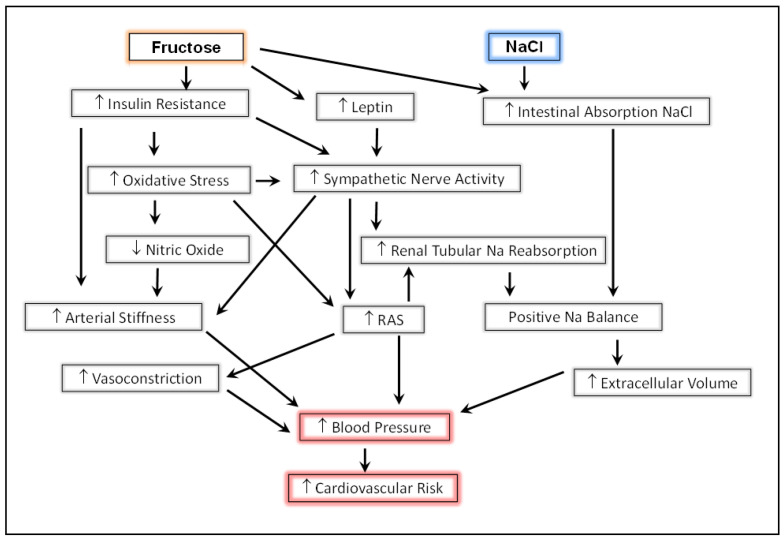
Interactions of pathways resulting in hypertension and increased cardiovascular risk induced by excess fructose and salt consumption (see text). RAS, renin angiotensin system.

**Table 1 nutrients-14-02515-t001:** Dietary sodium consumption in Asian populations.

No.	Author, Year	Country	*n*	Population	Method Dietary Assessment	Sodium Intakes or Sodium Excretion	Dietary Energy Intake
1	Lee, 2013 [50]	South Korea	9264	Subjects > 1 year old	24-h diet recall	Mean: 4.7 g	Not reported
2	Song, 2016 [51]	South Korea	9869	Adults (19–64 years)	Food Frequency Questionnaire + a 24-h diet recall	Mean sodium: 4943 mg/d Males: 6045 mg/d Females: 4240 mg/d	Energy: 1910 kcal/d Males: 2322 kcal/d Females:1648 kcal/d
3	Choi, 2018 [52]	South Korea	10,672	Adults > 18 years	24-h urine sodium	Mean (SE): 3.3 (0.1) g/d	Not reported
4	Kim, 2018 [53]	South Korea	718	Children & adolescents (10–18 years)	24-h urine sodium	Mean: 4190.55 mg	Mean 2270.70 kcal/d
5	Park, 2020 [54]	Korea	N/R	KNHANES 2014 (Subjects ≥ 1 year)	24-h diet recall	Total sodium: 3889.9 mg Male: 4557 mg Female: 3222.6 mg By age 1–2 years: 1111.7 mg 3–5 years: 1775.1 mg 6–11 years: 2916.5 mg 12–18 years: 3766.5 mg 19–29 years: 4226.4 mg 30–49 years: 4448.4 mg 50–64 years: 4161.4 mg 65+ years: 2987.1 mg	Not reported
6	Uechi, 2017 [55]	Japan	N/R	Systematic Review	24-h urine sodium	Mean (±SE) Overall: 4900 ± 190 mg Males: 5280 ± 310 mg Females: 4620 ± 290 mg	Not reported
7	Okuda, 2021 [56]	Japan	2377	Children (10–11 and 13–14 years)	Brief diet history questionnaire	Mean sodium: 4179 ± 1162 mg/d	Energy: 2005 ± 570 kcal/d
8	Wen, 2019 [57]	Japan China	1,145,839	Adults (40–59 years)	24-h diet recall	Japan: 202.2 ± 55.6 mmol/d China: 173.5 ± 84.5 mmol	Not reported
9	Tan, 2019 [58]	China	N/R	Systematic Review	Systematic Review	Mean urinary sodium excretion (range) 3–6 years: 86.99 (69.88, 104.1) mmol/24 h 6–16 years: 151.09 (131.55, 170.63) mmol/24 h ≥16 years: 189.07 (182.14, 195.99) mmol/24 h	Not reported
10	Powles, 2013 [59]	China Japan Korea	N/R	Systematic Review	Systematic Review	All values in g/d China: Total: 4.83 (4.62–5.05) Male: 5.05 (4.71–5.39) Female: 4.60 (4.31–4.88) Japan Total: 4.89 (4.71–5.08) Male: 5.12 (4.85–5.41) Female: 4.68 (4.43–4.93) Korea Total: 3.79 (3.16–4.46) Male: 4.01 (3.07–5.21) Female: 3.59 (2.79–4.55)	Not reported
11	Zhu, 2021 [60]	China (Shanghai)	3958	Adolescents (6–17 years)	Three 24-h dietary recalls	Dietary sodium intake 4297.6 ± 2285.5 mg/d Dietary source of sodium: salt (57.4%); soy sauce (13.2%), fungi and algae (6.5%) Mono Na glutamate (4.6%)	Not reported

**Table 2 nutrients-14-02515-t002:** Dietary fructose intake in Asian populations.

No.	Author, Year	Country	*n*	Population	Dietary Assessment Method	Dietary Fructose/Fructose-Rich Beverage Intakes
1	Takeichi, 2012 [77]	Japan	283	Children (7, 10 and 13 years)	Three 24-h dietary recalls	Fructose intake 7 year males: 3.5 ± 3.3 g/d 7 year females: 3.8 ± 3.6 g/d 10 year males: 3.4 ± 3.7 g/d 10 year females: 3.6 ± 2.8 g/d 13 year males: 4.4 ± 4.6 g/d 13 year females: 2.1 ± 2.7 g/d Overall: 3.5 ± 3.5 g/d
2	Wang, 2014 [78]	Japan	1631	Healthy control and individuals with colorectal cancer	Personal computer assisted dietary interview	Fructose intake Males (colorectal cancer): 6.17 (2.70–11.65) g/d Males (control): 6.11 (2.82–10.77) g/d Females (colorectal cancer): 8.98 (5.67–14.58) g/d Females (control): 9.63 (5.98–14.87) g/d
3	Fujiwara, 2018 [79]	Japan	2335	Toddlers (18–35 month) preschool children (3–6 years) schoolchildren (8–14 years) adults (20–69 years)	Weighted Dietary Record	Fructose intake Toddlers (Boys): 7.8 ± 4.3 g/d Toddlers (Girls): 7.6 ± 5.4 g/d Preschool children (Boys): 10.0 ± 3.7 g/d Preschool children (Girls): 9.3 ± 2.7 g/d Schoolchildren (Boys): 11.1 ± 3.8 g/d Schoolchildren (Girls): 10.3 ± 4.0 g/d Adults (Males): 10.1 ± 4.3 g/d Adults (Females): 9.9 ± 4.0 g/d
4	Yamakawa, 2020 [80]	Japan	22394	Adults (≥35 years)	Food Frequency Questionnaire	Fructose intake Males: 15.0 ± 10.0 g/d Females: 13.7 ± 8.1 g/d
5	Cho, 2020 [81]	Japan	1435	Healthy control and individuals with colorectal adenoma	Food Frequency Questionnaire	Fructose intake Colorectal adenoma (*n* = 738): 10.4 (7.0–14.8) g/d Control (*n* = 697): 11.1 (7.6–15.1) g/d
6	Edo, 2021 [82]	Japanese-American	555	Healthy control and individuals with early age-related macular degeneration (AMD)	Food Frequency Method	Fructose intake AMD (*n* = 111): 2.8 ± 3.6 g/d Control (*n* = 444): 3.7 ± 4.2 g/d
7	Shikanai, 2016 [83]	Taiwan	410	Children (7, 10 and 12 years)	Three 24-h dietary recalls	Fructose intake 7 year males: 7.3 ± 7.1 g/d 7 year females: 8.7 ± 6.9 g/d 10 year males: 9.5 ± 8.7 g/d 10 year females: 7.2 ± 6.6 g/d 12 year males: 8.5 ± 10.7 g/d 12 year females: 7.5 ± 7.4 g/d Overall: 8.8 ± 8.1 g/d
8	Pang, 2021 [84]	China	25528	Adults (≥45 years)	Three 24-h dietary records	Total fructose intake average (median) City: 11.6 (8.3) g/d Rural: 7.6 (5.3) g/d Free-fructose City: 6.7 (4.2) g/d Rural: 4.6 (2.7) g/d Bound-fructose intake City: 4.9 (3.7) g/d Rural: 3.1 0 (2.4) g/d

**Table 3 nutrients-14-02515-t003:** Fructose intake as sugar-sweetened beverages in Asian populations.

No.	Author, Year	Country	*n*	Population	Dietary Assessment Method	Dietary Fructose/Fructose-Rich Beverage Intakes	Dietary Energy Intake
1	Lee, 2013 [87]	South Korea	5421	Children & adolescent (7–18 years)	24-h dietary recall	SSB intake Overall: 7–18 year: 98.7 mL/d Males 7–18 year: 114.1 mL/d Females 7–18 year: 82.1 mL/d Overall: 7–12 year: 64.7 mL/d Overall: 13–18 year: 120.2 mL/d	Mean energy intake for all: 1991 kcal/day
2	Bae, 2014 [88]	Korea	9400	Apparently healthy adult (Korean Multi-Rural Communities Cohort Study)	Food Frequency Questionnaire	Soft drink intake Males (*n* = 3564): 16.2 ± 56.7 mL/d Females (*n* = 5836): 8.7 ± 45.0 mL/d	Energy intake Males (*n* = 3564): 1698.5 ± 494.1 mL/d Females (*n* = 5836): 1481.0 ± 437.0 mL/d
3	Ha, 2016 [89]	South Korea	2599	Children (9–14 years)	Dietary records 3–7 days	Mean percent energy from SSB Males: 5.8% Females: 6.0%	Mean energy intake: Males: 1846 kcal/day Females: 1617 kcal/day
4	Song, 2016 [90]	South Korea	9869	Adults (19–64 years)	Food Frequency Questionnaire + 24-h diet recall	Mean carbonated SSB Overall: 21.8 g/d Males: 28.9 g/d Females: 17.2 g/d	Energy intake: 1910 kcal/d Males: 2322 kcal/d Females: 1648 kcal/d
5	Lim, 2017 [91]	South Korea	49,826	Subject aged > 1 year	24-h dietary recall	SSB consumption per consumer as energy intake Year 1998: 123 kcal Year 2001: 141 kcal Year 2005: 126 kcal Year 2007–09: 166 kcal	Not reported
6	Kwak, 2018 [92]	South Korea	5775	Adult (≥40 years old)	Food Frequency Questionnaire	Mean SSB intake: 1.5 serving/week	Energy intake: 1938 kcal/d
7	Sakurai, 2013 [93]	Japan	2037	Adult men (35–44 years)	Diet History Questionnaire	Median (range) SSB intake: 0.2 (0.0–9.6) servings ^†^/d	Mean energy intake: 2194 kcal/d
8	Lin, 2016 [94]	Taiwan	1454	Adolescents (12–16 years)	Food Frequency Questionnaire	SSB intake Nonuser = 11% 1–350 mL/d = 38% 351–750 mL/d = 44% >750 mL/d = 7%	Energy intake (kcal/d) Nonuser: 1930 ± 52.7 1–350 mL/d: 1901 ± 27.9 351–750 mL/d: 2194 ± 42 >750 mL/d: 2425 ± 151
9	Li, 2014 [95]	China	29,215	Adult (≥18 years old)	Not reported	SSB consumption rate 2004: 15.1% 2006: 14.9% 2009: 29.3%	Not reported
10	Gui, 2017 [96]	China	53,151	Adolescents (6–17 years)	Self-reported dietary questionnaire	SSB per capita: 0.41 serving */d SSB per consumer: 0.61 serving */d	Not reported
11	Li, 2020 [97]	China	5,258	Adolescents (7–18 years)	Simplified self-reported dietary questionnaire	SSB intake Non-SSB drinker = 33.4% 0–0.3 serving */d = 36.8% >0.3 serving */d = 29.8%	Not reported
12	Gan, 2021 [98]	China	25,553	Adolescents (6–17 years)	Food Frequency Questionnaire	SSB intake: 181 g/d 6–10 year old: 129.5 g/d 11–14 year old: 208 g/d 15–17 year old: 285 g/d	Median energy intake: 1892.3 (1419.7–2572.6) kcal/d
13	Gui, 2021 [99]	China	6,387	Children (6–12 years)	Self-reported dietary questionnaire	Mean SSB intake per week per SSB consumer: 2.45 servings *	Not reported
14	Zhu, 2021 [100]	China (Shanghai)	3958	Adolescents (6–17 years)	Three 24-h dietary recalls	SSB consumption 59.4 ± 126.3 g/d	Not reported

* 1 serving = 250 mL SSB, ^†^ 1 serving = 237 mL SSB. Abbreviation: SSB, sugar-sweetened beverages.

## Data Availability

The data in this review were obtained using the following databases: PubMed https://www.ncbi.nlm.nih.gov/pubmed/ (accessed 2 June 2022); the Web of Science Core Collection https://www.library.ethz.ch/en/Resources/Databases/Web-of-Science-Core-Collection (accessed on 25 April 2022); and the Cochrane Registry http://www.cochranelibrary.com/about/central-landing-page.html (accessed on 25 April 2022).

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
