# Peer review of "Impact of Dietary Fructose and High Salt Diet: Are Preclinical Studies Relevant to Asian Societies?"

_nutrients, 2022, doi:10.3390/nu14122515_

Round 1
Reviewer 1 Report
The authors of this article entitled ‘‘Impact of dietary fructose and high salt diet: Are preclinical studies relevant to Asian societies?’’ aimed to review the dietary habits of Asian populations and assess whether existing preclinical data regarding fructose and salt consumption may be relevant to East Asian societies. As highlighted by the authors, this review is not meant to be exhaustive. The purpose is to provide insight into the similarities and differences of dietary patterns across different nations, so that future preclinical and clinical trials can be modeled accordingly, by incorporating all the available relevant data. Overall it is a well written and organized manuscript. Below, several remarks are presented to the authors.
1. At the end of the materials and methods section, it would be nice to mention, if possible, the number of relevant human and animal studies of fructose and salt consumption pertinent to East Asian dietary patterns, which were finally retrieved by your search, so as to highlight the paucity of exisiting data on the subject.
2. When using abbreviations and acronyms, they should first be presented in the expanded form and abbreviated thereafter. e.g. PKC (Line 199), SE (Table 1), RAS (Figure 1)
3. Lines 82-85: ‘‘A recent cohort study conducted in northern China found that among 2057 participants recruited to date`, SSBP prevalence is the same between hypertensive (28.5%) and normotensive (28.2%) groups`, with daily salt intake being higher in the former (8.97 grams/day) compared to the latter (6.87 grams/day) [35].’’ Please provide correct reference, since the reported statement is not supported by reference 35.
4. Lines 108-109: Figures provided here ‘‘2.3 g/d sodium intake (equivalent to 5.85 g/d salt) endorsed by the American College of Cardiology and American Heart Association [50]’’ should correspond to the figures mentioned earlier in Lines 86-87 ‘‘the upper limit recommended by the American Heart Association, which is 1,500 mg of sodium/day, which equates to about 3.25 g of salt/day [36].’’ Please correct accordingly using the appropriate references and the latest guidelines.
5. Lines 151-154: Please provide appropriate reference for the following sentence ‘‘Results from the large western CARDIA (Coronary Artery Risk Development in Young Adults; n = 240,508) cohort indicate that the risk of hypertension with consumption of SSB is 12% higher even when controlled for sex, age, race, BMI and smoking history [64].’’ I am not sure if you are referring to the following metaanalysis: «Jayalath VH, de Souza RJ, Ha V, Mirrahimi A, Blanco-Mejia S, Di Buono M, Jenkins AL, Leiter LA, Wolever TM, Beyene J, Kendall CW, Jenkins DJ, Sievenpiper JL. Sugar-sweetened beverage consumption and incident hypertension: a systematic review and meta-analysis of prospective cohorts. Am J Clin Nutr. 2015;102(4):914-21.»
6. Lines 151-156: Here there is lack of coherency. The following sentences should be placed in another part of section 3.3 or in a different paragraph, since this paragraph starting from Line 138 is referring to studies including Asian populations: ‘‘Results from the large western CARDIA (Coronary Artery Risk Development in Young Adults; n = 240,508) cohort indicate that the risk of hypertension with consumption of SSB is 12% higher even when controlled for sex, age, race, BMI and smoking history [64]. Comparable large clinical trials or epidemiological studies regarding the impact of fructose either with or without salt modulation on blood pressure in Asian populations are lacking and are very much needed.’’
7. Lines 188-191: ‘‘The putative anion transporter-1 (PAT1, also known as SCL26A6 in humans) is a chloride/base exchanger on the apical surface of intestinal cells. Jejunal PAT1 colocalizes with GLUT5; PAT1 increases with fructose feeding and is coupled with the intestinal Na/H exchanger-3 (NHE3) [74,77-79].’’ Reference 78 is not considered relevant to the above statements.
8. Lines 198-201: ‘‘Likewise, the activity of proximal tubule Na/H exchanger, NHE3, isolated from rats fed 20% fructose in drinking water is augmented in a PKC-dependent manner when glucose is replaced with fructose in the test bath. Significantly, systolic blood pressure was 13 mmHg higher in the fructose-fed vs glucose-fed rats [81].’’. It would be preferable to use the same tense in all sentences referring to the same experiment. This makes it easier for the reader to comprehend and keep up with the writing flow. Ideally you should use past tense when referring to certain experiments (regardless the tense you use, the same writing style should be kept all throughout the text or at least throughout the same paragraph).
9. Lines 200-201: ‘‘Significantly, systolic blood pressure was 13 mmHg higher in the fructose-fed vs glucose-fed rats [81].’’ Please revise the sentence, since according to reference 81 the BP difference was observed between fructose-fed rats and rats fed on a combination of fructose and high salt diet (not glucose-fed rats).
10. There are some expression and editing issues. Certain parts of the manuscript may need some amendments in order to become more reader friendly and easily comprehensible.
11. Several commas throughout the text are redundant. I would suggest not using Oxford comma, since it can sometimes lead to confusion and negatively affect the normal flow of reading. The use of Oxford comma can be found throughout the text i.e. Line 33 (China,) Line 91 (synthase [39], and) Line 95 (genes,) Line 114 (Kingdom,) Line 120 (pressure,) Line 226 (dysfunction) Line 227 (oxide,) Line 235 (system,) Line 237 (sympathoexcitation,)
12. Lines 12-13: Please rephrase the following phrase ‘‘other deleterious cardiovascular parameters’’. Instead of the word parameters, I would suggest using another word, such as consequences, diseases, entities or disorders.
13. Line 33: Please correct the word ‘‘This aim’’ to ‘‘The aim’’.
14. Line 37: The word ‘‘and’’ should be removed from Line 37 and placed in Line 39 (i.e. ‘‘and the Cochrane Registry’’)
15. Lines 47-50: For purposes of better clarity, please revise the sentence contained in these lines. Could it be that the meaning of the sentence is as follows ‘‘Relevant human and animal studies were included that addressed fructose and salt consumption in East Asian populations, as well as reviews regarding use of these nutrients by western countries for comparison, and animal studies, primarily in rodents, on fructose and high salt diet.’’?
16. Lines 55-56: Please rephrase the sentence ‘‘The factors influencing one’s susceptibility to high blood pressure may be related to genetic [20] or lifestyle factors [10].’’ so that it doesn’t contain the word ‘‘factors’’ twice. Possible suggestion: genetic or lifestyle characteristics.
17. Line 62: Please replace period (.) with comma (,) after the word ‘‘Europeans’’.
18. Lines 65-67: For purposes of better writing flow, I would suggest adding a phrase which separates the first from the second half of the sentence, so that the reader can easily comprehend that a second sentence starts after the word ‘‘susceptibilities’’ (e.g one can argue that lifestyle factors…). I would also put a comma after the word ‘‘diet’’ in Line 67.
19. Line 72: Please revise the phrase ‘‘and that remain to be identified’’.
20. Lines 77-78: Please revise the sentence ‘‘Interestingly, SSBP is an independent risk factor for cardiovascular disease independent of hypertension status [33].’’
21. Line 87: Please revise, since 1,500 mg of sodium/day equates to about 3.75 g of salt/day (and not 3.25)
22. Lines 89-90: Please rephrase the sentence ‘‘These occur in the genes encoding the proteins involved in pathways regulating salt balance and include genes encoding for angiotensinogen…’’
23. Lines 93-96: Please revise the sentence ‘‘The GenSalt study identified genetic associations potassium-selective ion channels (Kir), cyclic GMP-dependent protein kinase genes, and the epithelial sodium channel (ENaC) in among participants in rural China.’’
24. Line 111: Please correct ‘‘parameter’’ to ‘‘parameters’’.
25. Line 113: For better clarity, I believe that you should specify that you are referring to sodium intake when talking about dietary recall.
26. Line 123: Please revise the phrase ‘‘the association with fructose and obesity and cardiovascular disease’’
27. Line 128: Please correct ‘‘wester’’ to ‘‘western’’.
28. Lines 130-131: Please correct the phrase ‘‘suggests that intake as tended to increase’’
29.Lines 133-134: Please correct the phrase ‘‘such as absolute caloric intake per day vs as percent of total caloric intake’’ by removing ‘‘as’’ after vs.
30. Lines 135-136: Please revise the sentence ‘‘The problem is compounded by different definitions of that constitute a serving of SSB, not to mention…’’.
31. Lines 138-139 and 139-141: I would suggest adding commas to the sentences as follows: ‘‘There is a paucity of data regarding the impact, if any, by fructose intake, either alone or together with dietary sodium, on blood pressure…’’ and ‘‘The INTERMAP study [55,61], that examined the relationship of blood pressure as a major risk factor for cardiovascular disease, included individuals…’’
32. Lines 144-146: Please revise the sentence ‘‘One randomized, double-blind, cross-over trial enrolled 18 healthy young Chinese adults of both sexes examined the effect of drinking…’’
33. Line 168: I would suggest revising the phrase ‘‘rats fed 20% fructose w/w’’ to ‘‘rats fed on 20% fructose w/w’’. The same should be applied to Lines 172 (rats fed fructose), 173 (rats fed fructose), 199 (rats fed 20% fructose), 213 (rats fed 20% fructose) and 214 (rats fed fructose).
34. Line 183-184: ‘‘which has low affinity for glucose by high affinity for fructose’’. Please correct.
35. Lines 186-187: ‘‘where fructose can then enhance both intestinal sodium absorption and increase renal tubule sodium reabsorption’’. Please revise to ‘‘where fructose can then both enhance intestinal sodium absorption and increase renal tubule sodium reabsorption’’
Author Response
We wish to thank the reviewers for their thorough review and excellent suggestions regarding accuracy of our statements and interpretations. We also appreciate the helpful comments on usage as we were under a very tight deadline and many grammatical and other issues slipped by. We have attempted to address each concern.
- At the end of the materials and methods section, it would be nice to mention, if possible, the number of relevant human and animal studies of fructose and salt consumption pertinent to East Asian dietary patterns, which were finally retrieved by your search, so as to highlight the paucity of existing data on the subject.
Done.
- When using abbreviations and acronyms, they should first be presented in the expanded form and abbreviated thereafter. e.g. PKC (Line 199), SE (Table 1), RAS (Figure 1)
Done except for SE, standard error (This is a standard abbreviation for statistics).
- Lines 82-85: ‘‘A recent cohort study conducted in northern China found that among 2057 participants recruited to date`, SSBP prevalence is the same between hypertensive (28.5%) and normotensive (28.2%) groups`, with daily salt intake being higher in the former (8.97 grams/day) compared to the latter (6.87 grams/day) [35].’’ Please provide correct reference, since the reported statement is not supported by reference 35.
This was a reference by Qi that EndNote decided to substitute! Have typed it in separately.
- Lines 108-109: Figures provided here ‘‘2.3 g/d sodium intake (equivalent to 5.85 g/d salt) endorsed by the American College of Cardiology and American Heart Association [50]’’ should correspond to the figures mentioned earlier in Lines 86-87 ‘‘the upper limit recommended by the American Heart Association, which is 1,500 mg of sodium/day, which equates to about 3.25 g of salt/day [36].’’ Please correct accordingly using the appropriate references and the latest guidelines.
The latest guidelines are for the higher amount of salt from 2019. A comment has been made regarding this difference in recommendations. Thank you for catching the discrepancy.
- Lines 151-154: Please provide appropriate reference for the following sentence ‘‘Results from the large western CARDIA (Coronary Artery Risk Development in Young Adults; n = 240,508) cohort indicate that the risk of hypertension with consumption of SSB is 12% higher even when controlled for sex, age, race, BMI and smoking history [64].’’ I am not sure if you are referring to the following metaanalysis: «Jayalath VH, de Souza RJ, Ha V, Mirrahimi A, Blanco-Mejia S, Di Buono M, Jenkins AL, Leiter LA, Wolever TM, Beyene J, Kendall CW, Jenkins DJ, Sievenpiper JL. Sugar-sweetened beverage consumption and incident hypertension: a systematic review and meta-analysis of prospective cohorts. Am J Clin Nutr. 2015;102(4):914-21.»
Thanks for catching this. Done.
- Lines 151-156: Here there is lack of coherency. The following sentences should be placed in another part of section 3.3 or in a different paragraph, since this paragraph starting from Line 138 is referring to studies including Asian populations: ‘‘Results from the large western CARDIA (Coronary Artery Risk Development in Young Adults; n = 240,508) cohort indicate that the risk of hypertension with consumption of SSB is 12% higher even when controlled for sex, age, race, BMI and smoking history [64]. Comparable large clinical trials or epidemiological studies regarding the impact of fructose either with or without salt modulation on blood pressure in Asian populations are lacking and are very much needed.’’
Done.
- Lines 188-191: ‘‘The putative anion transporter-1 (PAT1, also known as SCL26A6 in humans) is a chloride/base exchanger on the apical surface of intestinal cells. Jejunal PAT1 colocalizes with GLUT5; PAT1 increases with fructose feeding and is coupled with the intestinal Na/H exchanger-3 (NHE3) [74,77-79].’’ Reference 78 is not considered relevant to the above statements.
Done.
- Lines 198-201: ‘‘Likewise, the activity of proximal tubule Na/H exchanger, NHE3, isolated from rats fed 20% fructose in drinking water is augmented in a PKC-dependent manner when glucose is replaced with fructose in the test bath. Significantly, systolic blood pressure was 13 mmHg higher in the fructose-fed vs glucose-fed rats [81].’’. It would be preferable to use the same tense in all sentences referring to the same experiment. This makes it easier for the reader to comprehend and keep up with the writing flow. Ideally you should use past tense when referring to certain experiments (regardless the tense you use, the same writing style should be kept all throughout the text or at least throughout the same paragraph).
Again, thank you. Done.
- Lines 200-201: ‘‘Significantly, systolic blood pressure was 13 mmHg higher in the fructose-fed vs glucose-fed rats [81].’’ Please revise the sentence, since according to reference 81 the BP difference was observed between fructose-fed rats and rats fed on a combination of fructose and high salt diet (not glucose-fed rats).
Corrected.
- There are some expression and editing issues. Certain parts of the manuscript may need some amendments in order to become more reader friendly and easily comprehensible.
Have tried to improve readability.
- Several commas throughout the text are redundant. I would suggest not using Oxford comma, since it can sometimes lead to confusion and negatively affect the normal flow of reading. The use of Oxford comma can be found throughout the text i.e. Line 33 (China,) Line 91 (synthase [39], and) Line 95 (genes,) Line 114 (Kingdom,) Line 120 (pressure,) Line 226 (dysfunction) Line 227 (oxide,) Line 235 (system,) Line 237 (sympathoexcitation,)
Unless it is a standard of the journal not to do so, we prefer to continue use of the Oxford comma.
- Lines 12-13: Please rephrase the following phrase ‘‘other deleterious cardiovascular parameters’’. Instead of the word parameters, I would suggest using another word, such as consequences, diseases, entities or disorders.
Done.
- Line 33: Please correct the word ‘‘This aim’’ to ‘‘The aim’’.
Done.
- Line 37: The word ‘‘and’’ should be removed from Line 37 and placed in Line 39 (i.e. ‘‘and the Cochrane Registry’’)
Done.
- Lines 47-50: For purposes of better clarity, please revise the sentence contained in these lines. Could it be that the meaning of the sentence is as follows ‘‘Relevant human and animal studies were included that addressed fructose and salt consumption in East Asian populations, as well as reviews regarding use of these nutrients by western countries for comparison, and animal studies, primarily in rodents, on fructose and high salt diet.’’?
Yes, the meaning would be better phrased. We have added clarifying language.
- Lines 55-56: Please rephrase the sentence ‘‘The factors influencing one’s susceptibility to high blood pressure may be related to genetic [20] or lifestyle factors [10].’’ so that it doesn’t contain the word ‘‘factors’’ twice. Possible suggestion: genetic or lifestyle characteristics.
Done.
- Line 62: Please replace period (.) with comma (,) after the word ‘‘Europeans’’.
Done.
- Lines 65-67: For purposes of better writing flow, I would suggest adding a phrase which separates the first from the second half of the sentence, so that the reader can easily comprehend that a second sentence starts after the word ‘‘susceptibilities’’ (e.g one can argue that lifestyle factors…). I would also put a comma after the word ‘‘diet’’ in Line 67.
Done.
- Line 72: Please revise the phrase ‘‘and that remain to be identified’’.
We have deleted that phrase entirely.
- Lines 77-78: Please revise the sentence ‘‘Interestingly, SSBP is an independent risk factor for cardiovascular disease independent of hypertension status [33].’’
Done with greater clarity (I hope).
- Line 87: Please revise, since 1,500 mg of sodium/day equates to about 3.75 g of salt/day (and not 3.25)
Done.
- Lines 89-90: Please rephrase the sentence ‘‘These occur in the genes encoding the proteins involved in pathways regulating salt balance and include genes encoding for angiotensinogen…’’
Rephrased into two sentences.
- Lines 93-96: Please revise the sentence ‘‘The GenSalt study identified genetic associations potassium-selective ion channels (Kir), cyclic GMP-dependent protein kinase genes, and the epithelial sodium channel (ENaC) in among participants in rural China.’’
Done.
- Line 111: Please correct ‘‘parameter’’ to ‘‘parameters’’.
Done.
- Line 113: For better clarity, I believe that you should specify that you are referring to sodium intake when talking about dietary recall.
Clarification added.
- Line 123: Please revise the phrase ‘‘the association with fructose and obesity and cardiovascular disease’’
Done.
- Line 128: Please correct ‘‘wester’’ to ‘‘western’’.
Done.
- Lines 130-131: Please correct the phrase ‘‘suggests that intake as tended to increase’’
We meant to say “has tended”; corrected.
- Lines 133-134: Please correct the phrase ‘‘such as absolute caloric intake per day vs as percent of total caloric intake’’ by removing ‘‘as’’ after vs.
Done.
- Lines 135-136: Please revise the sentence ‘‘The problem is compounded by different definitions of that constitute a serving of SSB, not to mention…’’.
Revised and clarified.
- Lines 138-139 and 139-141: I would suggest adding commas to the sentences as follows: ‘‘There is a paucity of data regarding the impact, if any, by fructose intake, either alone or together with dietary sodium, on blood pressure…’’ and ‘‘The INTERMAP study [55,61], that examined the relationship of blood pressure as a major risk factor for cardiovascular disease, included individuals…’’
We do not thing the added commas would help in this case.
- Lines 144-146: Please revise the sentence ‘‘One randomized, double-blind, cross-over trial enrolled 18 healthy young Chinese adults of both sexes examined the effect of drinking…’’
Revised.
- Line 168: I would suggest revising the phrase ‘‘rats fed 20% fructose w/w’’ to ‘‘rats fed on 20% fructose w/w’’. The same should be applied to Lines 172 (rats fed fructose), 173 (rats fed fructose), 199 (rats fed 20% fructose), 213 (rats fed 20% fructose) and 214 (rats fed fructose).
Done. We think we caught each instance.
- Line 183-184: ‘‘which has low affinity for glucose by high affinity for fructose’’. Please correct.
Corrected.
- Lines 186-187: ‘‘where fructose can then enhance both intestinal sodium absorption and increase renal tubule sodium reabsorption’’. Please revise to ‘‘where fructose can then both enhance intestinal sodium absorption and increase renal tubule sodium reabsorption’’
Done.
We wish to thank the reviewers for their thorough review and excellent suggestions regarding accuracy of our statements and interpretations. We also appreciate the helpful comments on usage as we were under a very tight deadline and many grammatical and other issues slipped by. We have attempted to address each concern.
- At the end of the materials and methods section, it would be nice to mention, if possible, the number of relevant human and animal studies of fructose and salt consumption pertinent to East Asian dietary patterns, which were finally retrieved by your search, so as to highlight the paucity of existing data on the subject.
Done.
- When using abbreviations and acronyms, they should first be presented in the expanded form and abbreviated thereafter. e.g. PKC (Line 199), SE (Table 1), RAS (Figure 1)
Done except for SE, standard error (This is a standard abbreviation for statistics).
- Lines 82-85: ‘‘A recent cohort study conducted in northern China found that among 2057 participants recruited to date`, SSBP prevalence is the same between hypertensive (28.5%) and normotensive (28.2%) groups`, with daily salt intake being higher in the former (8.97 grams/day) compared to the latter (6.87 grams/day) [35].’’ Please provide correct reference, since the reported statement is not supported by reference 35.
This was a reference by Qi that EndNote decided to substitute! Have typed it in separately.
- Lines 108-109: Figures provided here ‘‘2.3 g/d sodium intake (equivalent to 5.85 g/d salt) endorsed by the American College of Cardiology and American Heart Association [50]’’ should correspond to the figures mentioned earlier in Lines 86-87 ‘‘the upper limit recommended by the American Heart Association, which is 1,500 mg of sodium/day, which equates to about 3.25 g of salt/day [36].’’ Please correct accordingly using the appropriate references and the latest guidelines.
The latest guidelines are for the higher amount of salt from 2019. A comment has been made regarding this difference in recommendations. Thank you for catching the discrepancy.
- Lines 151-154: Please provide appropriate reference for the following sentence ‘‘Results from the large western CARDIA (Coronary Artery Risk Development in Young Adults; n = 240,508) cohort indicate that the risk of hypertension with consumption of SSB is 12% higher even when controlled for sex, age, race, BMI and smoking history [64].’’ I am not sure if you are referring to the following metaanalysis: «Jayalath VH, de Souza RJ, Ha V, Mirrahimi A, Blanco-Mejia S, Di Buono M, Jenkins AL, Leiter LA, Wolever TM, Beyene J, Kendall CW, Jenkins DJ, Sievenpiper JL. Sugar-sweetened beverage consumption and incident hypertension: a systematic review and meta-analysis of prospective cohorts. Am J Clin Nutr. 2015;102(4):914-21.»
Thanks for catching this. Done.
- Lines 151-156: Here there is lack of coherency. The following sentences should be placed in another part of section 3.3 or in a different paragraph, since this paragraph starting from Line 138 is referring to studies including Asian populations: ‘‘Results from the large western CARDIA (Coronary Artery Risk Development in Young Adults; n = 240,508) cohort indicate that the risk of hypertension with consumption of SSB is 12% higher even when controlled for sex, age, race, BMI and smoking history [64]. Comparable large clinical trials or epidemiological studies regarding the impact of fructose either with or without salt modulation on blood pressure in Asian populations are lacking and are very much needed.’’
Done.
- Lines 188-191: ‘‘The putative anion transporter-1 (PAT1, also known as SCL26A6 in humans) is a chloride/base exchanger on the apical surface of intestinal cells. Jejunal PAT1 colocalizes with GLUT5; PAT1 increases with fructose feeding and is coupled with the intestinal Na/H exchanger-3 (NHE3) [74,77-79].’’ Reference 78 is not considered relevant to the above statements.
Done.
- Lines 198-201: ‘‘Likewise, the activity of proximal tubule Na/H exchanger, NHE3, isolated from rats fed 20% fructose in drinking water is augmented in a PKC-dependent manner when glucose is replaced with fructose in the test bath. Significantly, systolic blood pressure was 13 mmHg higher in the fructose-fed vs glucose-fed rats [81].’’. It would be preferable to use the same tense in all sentences referring to the same experiment. This makes it easier for the reader to comprehend and keep up with the writing flow. Ideally you should use past tense when referring to certain experiments (regardless the tense you use, the same writing style should be kept all throughout the text or at least throughout the same paragraph).
Again, thank you. Done.
- Lines 200-201: ‘‘Significantly, systolic blood pressure was 13 mmHg higher in the fructose-fed vs glucose-fed rats [81].’’ Please revise the sentence, since according to reference 81 the BP difference was observed between fructose-fed rats and rats fed on a combination of fructose and high salt diet (not glucose-fed rats).
Corrected.
- There are some expression and editing issues. Certain parts of the manuscript may need some amendments in order to become more reader friendly and easily comprehensible.
Have tried to improve readability.
- Several commas throughout the text are redundant. I would suggest not using Oxford comma, since it can sometimes lead to confusion and negatively affect the normal flow of reading. The use of Oxford comma can be found throughout the text i.e. Line 33 (China,) Line 91 (synthase [39], and) Line 95 (genes,) Line 114 (Kingdom,) Line 120 (pressure,) Line 226 (dysfunction) Line 227 (oxide,) Line 235 (system,) Line 237 (sympathoexcitation,)
Unless it is a standard of the journal not to do so, we prefer to continue use of the Oxford comma.
- Lines 12-13: Please rephrase the following phrase ‘‘other deleterious cardiovascular parameters’’. Instead of the word parameters, I would suggest using another word, such as consequences, diseases, entities or disorders.
Done.
- Line 33: Please correct the word ‘‘This aim’’ to ‘‘The aim’’.
Done.
- Line 37: The word ‘‘and’’ should be removed from Line 37 and placed in Line 39 (i.e. ‘‘and the Cochrane Registry’’)
Done.
- Lines 47-50: For purposes of better clarity, please revise the sentence contained in these lines. Could it be that the meaning of the sentence is as follows ‘‘Relevant human and animal studies were included that addressed fructose and salt consumption in East Asian populations, as well as reviews regarding use of these nutrients by western countries for comparison, and animal studies, primarily in rodents, on fructose and high salt diet.’’?
Yes, the meaning would be better phrased. We have added clarifying language.
- Lines 55-56: Please rephrase the sentence ‘‘The factors influencing one’s susceptibility to high blood pressure may be related to genetic [20] or lifestyle factors [10].’’ so that it doesn’t contain the word ‘‘factors’’ twice. Possible suggestion: genetic or lifestyle characteristics.
Done.
- Line 62: Please replace period (.) with comma (,) after the word ‘‘Europeans’’.
Done.
- Lines 65-67: For purposes of better writing flow, I would suggest adding a phrase which separates the first from the second half of the sentence, so that the reader can easily comprehend that a second sentence starts after the word ‘‘susceptibilities’’ (e.g one can argue that lifestyle factors…). I would also put a comma after the word ‘‘diet’’ in Line 67.
Done.
- Line 72: Please revise the phrase ‘‘and that remain to be identified’’.
We have deleted that phrase entirely.
- Lines 77-78: Please revise the sentence ‘‘Interestingly, SSBP is an independent risk factor for cardiovascular disease independent of hypertension status [33].’’
Done with greater clarity (I hope).
- Line 87: Please revise, since 1,500 mg of sodium/day equates to about 3.75 g of salt/day (and not 3.25)
Done.
- Lines 89-90: Please rephrase the sentence ‘‘These occur in the genes encoding the proteins involved in pathways regulating salt balance and include genes encoding for angiotensinogen…’’
Rephrased into two sentences.
- Lines 93-96: Please revise the sentence ‘‘The GenSalt study identified genetic associations potassium-selective ion channels (Kir), cyclic GMP-dependent protein kinase genes, and the epithelial sodium channel (ENaC) in among participants in rural China.’’
Done.
- Line 111: Please correct ‘‘parameter’’ to ‘‘parameters’’.
Done.
- Line 113: For better clarity, I believe that you should specify that you are referring to sodium intake when talking about dietary recall.
Clarification added.
- Line 123: Please revise the phrase ‘‘the association with fructose and obesity and cardiovascular disease’’
Done.
- Line 128: Please correct ‘‘wester’’ to ‘‘western’’.
Done.
- Lines 130-131: Please correct the phrase ‘‘suggests that intake as tended to increase’’
We meant to say “has tended”; corrected.
- Lines 133-134: Please correct the phrase ‘‘such as absolute caloric intake per day vs as percent of total caloric intake’’ by removing ‘‘as’’ after vs.
Done.
- Lines 135-136: Please revise the sentence ‘‘The problem is compounded by different definitions of that constitute a serving of SSB, not to mention…’’.
Revised and clarified.
- Lines 138-139 and 139-141: I would suggest adding commas to the sentences as follows: ‘‘There is a paucity of data regarding the impact, if any, by fructose intake, either alone or together with dietary sodium, on blood pressure…’’ and ‘‘The INTERMAP study [55,61], that examined the relationship of blood pressure as a major risk factor for cardiovascular disease, included individuals…’’
We do not thing the added commas would help in this case.
- Lines 144-146: Please revise the sentence ‘‘One randomized, double-blind, cross-over trial enrolled 18 healthy young Chinese adults of both sexes examined the effect of drinking…’’
Revised.
- Line 168: I would suggest revising the phrase ‘‘rats fed 20% fructose w/w’’ to ‘‘rats fed on 20% fructose w/w’’. The same should be applied to Lines 172 (rats fed fructose), 173 (rats fed fructose), 199 (rats fed 20% fructose), 213 (rats fed 20% fructose) and 214 (rats fed fructose).
Done. We think we caught each instance.
- Line 183-184: ‘‘which has low affinity for glucose by high affinity for fructose’’. Please correct.
Corrected.
- Lines 186-187: ‘‘where fructose can then enhance both intestinal sodium absorption and increase renal tubule sodium reabsorption’’. Please revise to ‘‘where fructose can then both enhance intestinal sodium absorption and increase renal tubule sodium reabsorption’’
Done.
Reviewer 2 Report
The following review addresses a very important issue of salt and fructose consumption in a society mired in metabolic syndrome and associated diseases.
However, the information is sparse and could be better presented in a short article rather than in a review.
In addition, the manuscript has the following observations
Correct capitalization (line 62)
Specify genes associated with sugar and salt intake and their implication for metabolic syndrome.
Correct space before parentheses (line 83)
Why fructose and not sucrose?
Missing space (line 171)
Different font sizes (line 229)
It is too short to have more than 100 references and these are mostly outdated
Author Response
Reviewer 2
We wish to thank the reviewer for the critical appraisal of our review.
The following review addresses a very important issue of salt and fructose consumption in a society mired in metabolic syndrome and associated diseases.
However, the information is sparse and could be better presented in a short article rather than in a review.
We beg to differ. The information in the Tables is extensive and includes what is available re: the intake of salt and fructose and sugar sweetened beverages in this population. As noted in the revision, despite a broad search, there is a paucity of information on the impact of fructose and salt in East Asian populations. This is itself a reason for a review as a spur to having more and better designed studies to address this important problem. The references go back from 1980 to the present and rather than being “outdated” the sparseness of the available information is itself an important aspect of research in this area. Many more studies have focused on western-style diets to the detriment of evaluating the impact of pre-clinical and clinical studies on diets consumed by 1/3 of the world’s population.
In addition, the manuscript has the following observations
Correct capitalization (line 62)
Done.
Specify genes associated with sugar and salt intake and their implication for metabolic syndrome.
This review is focused on the impact of fructose and salt consumption on blood pressure. Although the influence of fructose on metabolic syndrome is well recognized, the role that these nutrients play in blood pressure regulation even independent or preceding the full expression of metabolic syndrome is not as well appreciated. We have included a statement about this aspect.
Correct space before parentheses (line 83)
Thank you for catching this. Corrected.
Why fructose and not sucrose?
The deleterious effects of fructose relate primarily to the levels of fructose that can be achieved in the bloodstream (both systemic and portal blood). Ingestion of fructose as sucrose or in natural fruit, does not increase plasma fructose levels as much as fructose in its more concentrated pure form ingested as high fructose corn syrup (Metabolism 61:641-651, 2012). In a report by the Mayo clinic evaluating 24 prospective cohort studies comprising > 600,000 unique individuals, total sugars and fructose were associated with cardiovascular mortality, but not incidence. Sucrose was not associated with either cardiovascular incidence or mortality (Mayo Clinic Proc 94:2399-2414, 2019).
Sucrose is a disaccharide formed by fructose and glucose. Sucrose with high salt can also promote elevations in blood pressure but greater amounts of sucrose are required to achieve the same levels of fructose. It is the fructose component that leads to hypertension as can be seen with controls using glucose as the sole sugar in pre-clinical tests.
Because fructose has a low glycemic index, that is, it raises serum glucose levels very little since it must first be converted by the liver to glucose. As a result, insulin secretion in response to fructose is also limited. In addition, leptin (a hormone that signals satiety) is lower, but ghrelin (a hormone signaling hunger) is higher with fructose intake. Thus, fructose consumption impairs the normal signaling of energy metabolism. When ingested as sucrose, a disaccharide consisting of fructose plus glucose, the breakdown into its components permits glucose to proceed with normal signaling of energy intake. This mitigates some of the effects of the fructose.
We have added a brief comment to address this in the review.
Missing space (line 171)
Done.
Different font sizes (line 229)
This is done in conformity with the template for Nutrients which reduces the font size in these sections.
It is too short to have more than 100 references and these are mostly outdated
The review is comprehensive, if not exhaustive. It is necessary to use “outdated” references for consumption data in order to demonstrate the changes in intake of fructose and salt over time. This is commented upon in the review and can be seen by scrutinizing the tables that are arranged chronologically as well as by country. The paucity of data regarding East Asian countries poses a further challenge to finding more contemporary references. Nonetheless, the references we have cited are needed to validate and support the statements made in our review.
Round 2
Reviewer 2 Report
The authors have adapted the bibliographic references and the focus of the study to the Asian continent. Taking into account the justification and criteria made by the authors on the observations of review 1, I recommend the publication of the work, which has effectively corroborated that the studies carried out on fructose and salt consumption are limited in the Asian continent.